# Physiological Assessment of the Health and Welfare of Domestic Cats—An Exploration of Factors Affecting Urinary Cortisol and Oxytocin

**DOI:** 10.3390/ani12233330

**Published:** 2022-11-28

**Authors:** Takumi Nagasawa, Yuichi Kimura, Koji Masuda, Hidehiko Uchiyama

**Affiliations:** 1Department of Human and Animal-Plant Relationships, Graduate School of Agriculture, Tokyo University of Agriculture, Funako 1737, Atsugi 243-0034, Kanagawa, Japan; 2Department of Animal Science, Faculty of Agriculture, Tokyo University of Agriculture, Funako 1737, Atsugi 243-0034, Kanagawa, Japan

**Keywords:** oxytocin, cortisol, domestic cat, physiological state

## Abstract

**Simple Summary:**

Cats are the most common pet animal worldwide. Therefore, enhancing their health and welfare is essential for owners. However, few studies have assessed the physiology of domestic cats noninvasively. In this study, we collected urine samples from domestic cats and quantified cortisol as an indicator of negative stress states, and oxytocin as a positive indicator. Furthermore, we investigated factors influencing these hormonal states, which reflect their physiological status. Our study revealed that the more frequently owners talked to and petted their cats, the higher their urinary oxytocin levels. We suggest that social interactions between cats and their owners are factors that may influence their health and welfare status.

**Abstract:**

Physiological samples are beneficial in assessing the health and welfare of cats. However, most studies have been conducted in specialized environments, such as shelters or laboratories, and have not focused on cats living in domestic settings. In addition, most studies have assessed physiological stress states in cats based on cortisol, and none have quantified positive indicators, such as oxytocin. Here, we collected urine samples from 49 domestic cats and quantified urinary cortisol, oxytocin, and creatinine using ELISA. To identify factors influencing hormone levels, owners responded to questionnaires regarding their housing environment, individual cat information, and the frequency of daily interactions with their cats. Using principal component analysis, principal component scores for daily interactions were extracted. These results showed that the frequency of tactile and auditory signal-based communication by owners was positively correlated with the mean concentration of oxytocin in the urine. Additionally, this communication was more frequent in younger cats or cats that had experienced a shorter length of cohabitation with the owner. However, no factors associated with urinary cortisol concentration were identified. Our study indicates that interactions and relationships with the owner influence the physiological status of cats and suggests that oxytocin is a valuable parameter for assessing their health and welfare.

## 1. Introduction

In Japan, approximately 8.95 million cats are kept in households, which is more than the number of dogs [1]. Few people perceive cats as mere pets and most consider them equivalent to family and friends [2]. In other words, owners recognize cats as special beings and form a special social bond with them. Cats have excellent social-cognitive abilities [3]. Furthermore, they can recognize human emotions (e.g., happiness and anger) from human communicative cues (e.g., the vocal tone and facial expressions of people) [4,5] and the state of attention and interest in themselves from a person’s gaze [6,7]. Previous studies have shown that cats increased ear and head movements upon hearing their owners’ voices, rather than the voices of others [8]. It has been suggested that cats discriminate between their owners and others, and show attention and interest in their owners.

Social interaction with owners is a factor for influencing cat welfare [9]. For example, contact with people reduces the risk of respiratory diseases in cats living in shelters [10,11,12]. Human interactions alter the physiological and psychological state of cats and decrease cortisol levels, which reflect their stress state and related behaviors [13]. However, individual differences exist in the social behavior of cats [6]. In addition, cats’ stress coping strategies are highly varied, and results vary depending on individual cat characteristics (e.g., gender, age, and personality traits) [9,13,14]. The experimental environment is also important. For example, in experiments conducted in environments such as shelters, environmental arrangements such as the installation of hiding boxes may decrease cat stress more than human interaction [15]. Furthermore, several studies have reported that social interactions with owners may trigger negative reactions in cats. If the cat has a sensitivity to petting [16] or if the owner handles the cat inappropriately [17], the cat’s aggression may increase. Cats also show negative behavioral responses when contacted by their owners rather than strangers [18]. Furthermore, owner personality may also influence the outcome of the experiment, as owners of older age and more neurotic tendencies have been found to adopt less favorable interactions with cats [19]. Thus, the relationship between social interactions with people and cat welfare is complex and still under debate.

Objective methods to quantify the physiological state of animals are useful in assessing their welfare [20,21]. For example, cortisol is a well-known physiological indicator because it can assess the activation of the hypothalamic-pituitary-adrenal axis [22]. Indeed, studies in cats have attempted to measure cortisol from blood [23,24], saliva [25,26], nail, and hair [27] samples. However, cortisol levels can vary due to factors such as an individual cat’s personality and diurnal variation [9]. Furthermore, the collection method also needs to be considered. Cats must be held and restrained when collecting physiological samples. The handling itself can induce stress, especially when blood or saliva is collected [22]. This could be an obstacle to conducting experiments on cats in the general household. In particular, cats do not undergo a transparent domestication process similar to dogs [28], and their highly independent behavioral characteristics may make it difficult for owners to train cats. Furthermore, because cats are solitary hunters and territorial animals [29], a visit by a researcher with handling skills may cause a stress response in the cat. One solution to these problems is to utilize physiological samples that allow noninvasive collection, such as feces [30,31,32] and urine [33,34,35,36,37,38]. The relationship between fecal cortisol levels and social interactions with owners in cats is not yet conclusive, as it may be influenced by factors such as the experimental environment and the cat’s personality [32,35,36]. However, the use of fecal samples has several advantages. Cats have the habit of urinating at a fixed location. Cat-specific toilets with urination trays take advantage of this habit and are widely used in households. These toilets are convenient for urinalysis in veterinary clinics and can also be used in research. In fact, several studies have been conducted on cats in households [31,32,35]. In addition, when urine is used, there is concern that the concentration may vary with urine volume, but this can be corrected using creatinine [36,38]. Although it is not possible to measure acute stress responses, it could be an advantage in assessing daily interactions between owners and cats on a daily basis. Another advantage of using urine to assess physiological status is that it allows the quantification of oxytocin.

Oxytocin has recently attracted attention as a positive indicator of animal health and welfare. It is a type of peptide hormone produced by the paraventricular and supraoptic nucleus of the hypothalamus, where it acts as a neurotransmitter in the central functions of the body [39]. Additionally, it is secreted into the blood from the posterior pituitary gland and is known to be involved in diverse peripheral functions [39]. Recent studies have measured oxytocin levels in domesticated animals, such as horses [40] and pigs [41]. In particular, studies have revealed that interactions between dogs and their owners increased oxytocin levels in both [42,43,44,45,46]. Some studies have quantified the oxytocin levels in the blood [47] and urine [36,38] of laboratory-housed cats, but not in household cats. Furthermore, in contrast to dogs, oxytocin levels in cats have been reported to increase when social interaction with their owners is restricted [36]. Additionally, oxytocin is an essential hormone involved in forming attachment relationships between mothers and their children [39,48]. For example, oxytocin concentrations at birth are proportional to the strength of the attachment relationship after birth [48], suggesting that oxytocin secretion is the basis for attachment formation between the human mother and child. Oxytocin can be the basis for investigating the social relationships between humans and animals as well as a useful indicator for assessing animal health and welfare [49]. Therefore, research focusing on oxytocin is needed in the field of cat–owner relationship studies [50].

In this study, we aim to assess the physiological status of cats living in general households. First, we quantify the oxytocin levels by collecting urine samples from cats living in general households noninvasively. We then investigate whether the housing environment, individual cat characteristics, and owner–cat relationship and communication influences urinary oxytocin and cortisol concentrations in cats. This is the first study to quantify the oxytocin levels in cats living in households.

## 2. Materials and Methods

### 2.1. Ethics Statement

All studies were conducted with the approval of the Ethics Committee for Laboratory Animals of the Tokyo University of Agriculture (Approval No. 180264) in accordance with the Helsinki Convention.

### 2.2. Test Animals

Thirty homes (n = 49; male cat = 24 and female cat = 25; mean age = 5.30 ± 4.06) recruited through social media participated in our study. To assess the physiological status of cats acclimated to the home environment, we included only those cats that were kept for more than 4 months [32].

### 2.3. Experimental Protocol

As we aimed to quantify the physiological state of cats under normal conditions, life events, such as visits to veterinary clinics, house construction, and the start of new animal care, were prohibited during the period of urine sample collection. In addition, urine collection instruments were sent to the owners for collecting the urine samples. All participants provided written informed consent.

### 2.4. Assay Methods

#### 2.4.1. Collection of Urine Sample

Owners collected cat urine noninvasively for a minimum of three days and used a dropper to dispense the urine collected in the lower tray of the two-layer cat plastic litter box. They collected urine as soon as they noticed their cats urinating, regardless of the time of day, and also recorded the time of urination (or the estimated time of urination) and the time at which it was stored. Urine was aliquoted into 15 mL centrifuge tubes and stored in a freezer at −20 °C before being delivered to the laboratory via refrigerated transport, and stored in lab freezers at −20 °C until further analysis could be conducted. On the day hormone concentrations were measured, urine samples were thawed and centrifuged at 3200 rpm for 15 min at 4 °C, and the supernatant fluid was used for measurements.

#### 2.4.2. Oxytocin

Oxytocin was pretreated to solid phase extraction using a Hyper Sep C18 column (3 mL/200 g, 608–303, Thermo Fisher Scientific, Waltham, MA, USA). Oxytocin concentrations were quantified using a DetectX^®^ Oxytocin Enzyme Immunoassay Kit (K048-H5, Arbor Assays LLC, Ann Arbor, MI, USA, goat anti-rabbit IgG). The standard curve ranged from 16.38 to 10,000 pg/mL, with a sensitivity and detection limit of 17.0 pg/mL and 22.9 pg/mL, respectively. The absorbance was measured at 450 nm using an iMark microplate reader (Bio-Rad, Tokyo, Japan). The intra-assay and inter-assay coefficient of variation (%CV) were 2.86% and 9.71%, respectively.

#### 2.4.3. Cortisol

Cortisol concentrations were quantified using a DetectX^®^ Cortisol Enzyme Immunoassay Kit (K003-H5W, Arbor Assays LLC, goat anti-mouse IgG). Urine samples were diluted 10-fold with assay buffer in the kit to prevent matrix interference and to maintain the calculated concentrations within the standard curve. The standard curve ranged from 50 to 3200 pg/mL, with a sensitivity and detection limit of 27.6 pg/mL and 45.4 pg/mL, respectively. Absorbance was measured at a wavelength of 450 nm. The intra-assay and inter-assay %CV were 2.74% and 19.58%, respectively.

#### 2.4.4. Creatinine

Creatinine concentrations were measured using the Jaffe method to correct variations in hormone concentrations due to urine volume. Oxytocin and cortisol concentrations were corrected by dividing them by the quantified creatinine concentration (cortisol: ng/mg creatinine; oxytocin: pg/mg creatinine).

### 2.5. Questionnaire

Owners responded to the cat housing environment (number of people in the household and number of cats), individual cat variables (age, length of cohabitation, and body weight), and frequency of daily interactions with their cats on a 5- and 7-point Likert scale (Table 1). The question group was divided into owner-initiated behavior (owner-initiated interaction: OI) and cat-initiated behavior (cat-initiated interaction: CI). The owner–cat attachment relationship was quantified using the Lexington attachment to pets scale (LAPS) [51]. When multiple cats from the same household participated in the experiment, owners responded with the individual cat variables and daily interactions for each cat.

The owners also answered the Feline Five questionnaire [52]. From the responses, we calculated five personality trait values for each cat: Neuroticism (such as insecure and anxious), Extraversion (such as decisive and smart), Dominance (such as bullying and dominant), Impulsiveness (such as impulsive and erratic), and Agreeableness (such as affectionate and friendly).

### 2.6. Physical Activity

To examine the effects of physical activity on the hormone levels of cats, we used Plus Cycle (JARMeC, Kanagawa, Japan) [53], a collar-mounted activity meter. Out of all the cats participating in our study, individuals for whom data could not be extracted due to machine malfunction or who exhibited a stress response (e.g., scratching the activity meter and trying to remove it) to the attachment of the activity meter were excluded from the analysis.

### 2.7. Statistical Analyses

All statistical analyses were performed using Bell Curve for Excel ver. 4.02 (Social Survey Research Information Co., Ltd., Tokyo, Japan). Statistical significance was set at *p* = 0.05. For each individual, mean concentrations of oxytocin and cortisol were calculated from the total urine samples collected for each individual in the study. We used the Mann–Whitney test to ascertain sex differences in oxytocin and cortisol concentrations and calculated effect sizes (r).

We calculated principal component scores by subjecting the frequency of interactions between owners and cats to principal component analysis (PCA). The goal of PCA in this study was not to clarify the potential principal component structure, but to create an overall synthetic variable by reducing variables. Therefore, we performed a PCA without rotation [54]. We aimed to generalize and quantify the daily interactions between cats and their owners as much as possible, and to examine the relationship between these interactions and the cat’s urinary hormone concentrations.

Spearman’s rank correlation coefficient (r_s_) was calculated to explore the influencing factors, such as principal component scores, housing environment, individual variables, physical activity, cat personality, and owner attachment on mean urinary oxytocin and cortisol concentrations.

## 3. Results

We collected 313 urine samples (6.39 ± 2.07 average samples/individual) from 49 cats (male = 24 and female = 25; mean age = 5.3 ± 4.06 years) to quantify the oxytocin and cortisol levels. Detailed information for each individual is provided in the Appendix A. Three cats that had mixed urine samples owing to multiple cats, two cats that were not spayed or neutered, and eight cats with urinary-related diseases, such as kidney disease, were excluded from the analysis. In addition, only samples frozen and stored within 60 min of urination were used for the oxytocin analysis [55]. Thus, five cats were excluded from the oxytocin analysis.

The mean urinary cortisol concentration of 36 cats was 3.83 ± 1.63 ng/mg creatinine. The mean urinary oxytocin concentration in 31 cats was 257.26 ± 126.86 pg/mg creatinine. There was no significant correlation between cortisol and oxytocin levels (r_s_ = 0.13, *p* > 0.05)

### 3.1. Individual and Environmental Variables

No sex differences existed in the mean concentrations of cortisol (U = 113, Z = 1.52, and r = 0.25, *p* > 0.05) and oxytocin (U = 79, Z = 1.60, and r = 0.29, *p* > 0.05). In addition, no significant correlation was observed between the mean hormone concentrations and environmental and individual variables (Table 2).

### 3.2. Cat Personality

No significant correlation was observed between the mean cortisol and oxytocin concentrations and cat personality (Table 3).

### 3.3. Owner–Cat Interactions and Owner Attachment

The criteria for using the principal components in the analysis were eigenvalues greater than 1 and a cumulative contribution of more than 60%. Three principal components were extracted from the OI questionnaire items (hereafter referred to as OI-PC1: tactile and auditory interaction, OI-PC2: playing and brushing interaction, and OI-PC3: food interaction, Table 4), and four principal components were extracted from the CI questionnaire items (hereafter referred to as CI-PC1: active interaction, CI-PC2: passive interaction, CI-PC3: ambivalent interaction, and CI-PC4: sleep interaction, Table 5).

Cortisol concentrations were not significantly correlated with principal component scores or owner attachment; however, there was a significantly positive correlation between the oxytocin concentrations and OI-PC1 (Table 6).

### 3.4. OI-PC1

OI-PC1 was positively correlated with scores for Impulsiveness and negatively correlated with the length of cohabitation and cat age (Table 7).

## 4. Discussion

We found that owner-initiated interactions with cats, primarily through touch and voice, were positively correlated with the mean oxytocin concentrations (Table 6). Interactions via touch and voice have been widely employed as question items in scales that quantify cat–human relationships [56,57] and are standard communication methods that many owners use with their cats. Our results suggest that many owners influence the physiological status of their cats through daily communication. Cats engage in allogrooming and allorubbing behaviors with other cats to strengthen social bonds. Similarly, they engage in these behaviors with humans [58]. They can also identify the voices of their owners [8], suggesting that they are extremely sensitive to vocal stimuli. Thus, it is suggested that tactile and auditory interactions initiated by the owner are stimuli that can influence the physiological state of the cat.

Our study focused on inter- rather than intra-individual variation in urinary hormones. Thus, the direction of causality between contact or vocal interactions and oxytocin concentrations remains unclear. Therefore, we discuss our results in two ways. First, we present the hypothesis that contact and vocal communication initiated by owners are factors that increase oxytocin levels in cats. For example, contact interactions from human parents are known to increase oxytocin levels in human infants, with associated health-promoting effects, such as lower stress and increased well-being [59,60]. Furthermore, vocal communication by human mothers to their human preterm infants decreases pain and increases oxytocin levels, thereby proving its importance [61]. The relationship between cats and their owners is frequently likened to that between a parent and child [2,51,56]. On the other hand, no conclusion has been reached as to whether cats, similar to human infants, form attachment relationships with their owners [62,63,64]. Some cats also exhibit separation anxiety-related behaviors toward their owners [65], and being petted by people decreases stress in cats [10,11,12,13]. However, the results of these studies are also under debate, as they may depend on factors such as an individual cat’s personality and the experimental environment [10,11,12,13]. The present study opens a new topic for discussion because it shows that social interaction initiated by owners may positively influence oxytocin levels in cats living in the general household.

Second, we discuss the hypothesis that higher oxytocin concentrations in cats facilitate the frequency of contact and vocal communication with their owners, which could be interpreted as a tendency of cats with higher urinary oxytocin concentrations to tolerate more contact communication with their owners. Our study exhibited that younger cats and shorter periods of owner–cat cohabitation were associated with greater frequency of owner-initiated contact and vocal interactions (Table 7). Domesticated animals have undergone numerous changes (external appearance, such as coat color, and temperament, fear, and aggression toward people) from their wild origin [66,67]. Kikusui et al. stated that cortisol and oxytocin play fundamental roles in the relationship between humans and dogs [68]. In particular, human–dog interactions increase oxytocin levels in both parties only in dogs, not wolves [43]. In other words, the influence of dogs and their owners on each other’s oxytocin concentration is the mechanism that forms a social relationship between them [69,70]. Consistent with the results of the above studies, this study showed that interaction initiated by owners increased urinary oxytocin levels in cats. In addition, the oxytocin receptor gene in cats associated with social behavior and personality differed between purebred and mongrel cats, with mixed cats having higher friendliness scores [71]. Thus, it is suggested that, as in dogs, the oxytocin system may be involved in the physiological mechanism of cat domestication. Indeed, several studies have reported that cats form attachment relationships with their owners [63,64]. However, opposite results have also been reported. For example, the increase in cats’ urinary oxytocin levels under experimental conditions in which social interaction with people is restricted [36]. In addition, cats and their owners do not form attachment relationships [62]. Therefore, the relationship between physiological aspects, such as oxytocin, in cats and their social relationships with their owners needs to be carefully discussed.

We observed that owners communicated via touch and voice with highly impulsive cats (Table 7). The causal relationship between these results is unknown and multiple hypotheses can be formulated. For example, it is possible that the behavior of highly impulsive cats elicits positive interactions with their owners. The variables comprising impulsivity scores are erratic and unpredictable [52]. In other words, cats with high impulsivity tend to engage in unexpected and highly unpredictable behavior toward their owners. It is possible that these behaviors drew the attention and interest of the owners, resulting in more contact and vocal communication. Nevertheless, there was no direct relationship between personality tendencies and cortisol or oxytocin levels in cats in our study. There is a link between a cat’s personality and stress response in cats [72,73]. However, several studies have reported no association between cortisol concentrations and personality traits in cats [30,31,32], which is consistent with the results of our study. One reason for the lack of a direct relationship between hormone levels and personality is the difficulty in quantifying feline personality traits. For example, Feline Five is a questionnaire in which owners subjectively respond. However, methods have been developed to objectively evaluate cats by observing their behavior, such as the feline temperament profile [25]. Future research should analyze the relationship between hormone levels and personality by developing more accurate methods to assess the personality of cats.

Cortisol has been used in many studies as an indicator of stress status in animals, including cats [74]. However, we failed to identify the factors associated with cortisol levels in cats. Several limitations exist in the use of cortisol to assess stress in cats. For example, cortisol levels fluctuate within the day [9]. In addition, the state of a cat’s cortisol concentration does not necessarily correlate with scores on behavioral indices [13,20], but it can form a negative correlation with hiding behavior [25] and vocalizations [33]. Therefore, when assessing stress in cats, it is necessary to evaluate not only physiological but also behavioral indicators [9]. Additionally, our study was conducted in a relaxed home environment, whereas most studies on cortisol have been conducted on cats living in environments such as shelters or laboratories. In particular, it can be assumed that cats living in shelters experience intense stress due to changes in their housing environments, which is reflected in their cortisol levels. Therefore, evaluation of cortisol is a valuable method for such studies. However, we aimed to evaluate the normal physiological state of cats, and owners collected the urine samples during periods when stress-inducing life events, such as environmental changes, did not occur. In other words, the experimental conditions were such that stress, as reflected in cortisol levels, was difficult to generate in the first place. In contrast, oxytocin acts as a diverse anti-stress agent in cortisol suppression and is associated with psychological well-being and social behavior [75]. Oxytocin is an indicator that allows us to observe the effects of factors other than stress. Thus, it is a more useful biomarker than cortisol for assessing the health and welfare of cats living in households.

This study has several limitations. First, it is difficult to identify causal relationships between physiological states and effect factors when urine is used as an assessment sample. Cats are animals that urinate multiple times daily. In other words, the urine sample reflects the physiological state of the cat on a time-unit basis [76]. For example, physiological samples, such as blood and saliva, reflect short-term conditions on a minute-by-minute basis; therefore, a causal relationship between hormonal fluctuations and the effect factors can be seen by observing behavioral data immediately prior to sample collection. Future research should utilize samples other than urine for more detailed causal identification. However, the collection of saliva, especially blood, requires invasive procedures, and an experimenter with specific skills to visit the household. There is also a concern that the visit and sample collection may induce a stress response in cats, thereby hindering the achievement of the study’s aims. Therefore, targeting urine samples that can be collected noninvasively by the owners themselves would be beneficial in assessing the natural physiological state of cats living in general households.

The second limitation was the difficulty in quantifying the attachment relationship between the owners and cats. While oxytocin has been shown to contribute to the formation of attachment relationships between owners and dogs, in cats, the role of oxytocin in developing attachment relationships remains unclear [42,43,44,45,46]. In this study, we could not demonstrate a relationship between the attachment between owners and cats and oxytocin levels in cats. One reason for this is the characteristics of the scale, which quantifies the attachment level. We used the LAPS to quantify the pet attachment levels of owners. We also quantified the Dominance score, which is associated with the owner–cat attachment [52]. However, the owners answered both indicators subjectively and not objectively. For example, there are multiple attachment styles, such as secure, insecure-ambivalent, and insecure-avoidant [63]. To evaluate these attachment styles, it is necessary to record the actual situations in which cats and owners communicate and analyze their behavior. Future research should identify attachment styles between the cats and their owners based on behavioral analysis and explore their relationship with oxytocin.

## 5. Conclusions

In conclusion, urinary oxytocin levels in cats living in general households are associated with the frequency of contact and vocal communication from their owners. In other words, our results suggest that social interactions with owners are an important factor influencing the physiological status of cats. Additionally, contact and vocal communication was more frequent in younger cats or cats that experienced a shorter length of cohabitation with their owners. This indicates that oxytocin secretion via communication might play an important role in the early stages of the formation of the owner–cat relationship. However, no factors that were associated with cortisol levels were identified, suggesting that oxytocin may be a valuable indicator for the physiological assessment of cats living in a non-stressful environment, such as a general household.

## Figures and Tables

**Table 1 animals-12-03330-t001:** List of questionnaire items and response alternatives.

Questionnaire Items	Initiate	Response Alternatives
Do you pet your cat?	Owner	1: I always do 2: Frequently 3: Sometimes, I do 4: I seldom do 5: Not at all
Do you hug your cat?	Owner	
Do you talk to your cat?	Owner	
Do you call your cat’s name?	Owner	
Does your cat rub its body or tail on you?	Cat	
Does your cat spend time near people throughout the day?	Cat	
Does your cat purr?	Cat	
Does your cat call to demand food?	Cat	
Does your cat call for something other than food?	Cat	
Does your cat call to get attention from family members?	Cat	
Does your cat respond when you call its name?	Cat	
Does your cat make eye contact with you?	Cat	
How often do you brush your cat?	Owner	1: At least twice a day 2: Once a day 3: 3 or 4 times a week 4: Once a week 5: Once a month 6: Once a year 7: I do not
Do you give your cat treats?	Owner	
How often do you invite your cat to play?	Owner	
Does your cat knead?	Cat	
Does your cat sit on the lap of a family member?	Cat	
Does your cat sleep with someone in the family?	Cat	

**Table 2 animals-12-03330-t002:** Correlation between hormone concentrations and individual and environmental variables.

Individual and Environmental Variables	Cortisol (ng/mg Creatinine)	Oxytocin (pg/mg Creatinine)
	*n* = 36	*n* = 31
	r_s_	r_s_
Age (years)	−0.14	−0.06
Body weight (kg)	0.15	0.05
Physical activity ^1^	0.28	0.16
Number of cats	0.18	−0.04
Number of family members	−0.15	−0.12
Length of cohabitation (years)	−0.28	−0.16

^1^ For Physical activity, the number of samples (*n*) for cortisol and oxytocin were 24 and 28, respectively.

**Table 3 animals-12-03330-t003:** Correlation between hormone levels and cat personality.

Cat Personality	Cortisol (ng/mg Creatinine)	Oxytocin (pg/mg Creatinine)
	*n* = 36	*n* = 31
	r_s_	r_s_
Neuroticism	0.03	−0.18
Extraversion	0.07	0.09
Dominance	0.14	−0.21
Impulsiveness	−0.02	0.04
Agreeableness	−0.24	0.17

**Table 4 animals-12-03330-t004:** Principal component loadings and contributions to owner-initiated interaction.

Questionnaire Items	OI-PC1	OI-PC2	OI-PC3
Do you talk to your cat?	**0.92**	−0.13	0.01
Do you pet your cat?	**0.79**	0.20	−0.28
Do you hug your cat?	**0.73**	0.38	−0.21
Do you call your cat’s name?	**0.73**	−0.21	0.29
How often do you invite your cat to play?	−0.10	**0.77**	0.27
How often do you brush your cat?	−0.07	**0.76**	0.19
Do you give your cat treats?	0.22	−0.18	**0.89**
Eigenvalue	2.60	1.56	1.20
Contribution rate (%)	32.55	19.48	14.97
Cumulative contribution rate (%)	66.99		

Loads above |0.50|are shown in bold.

**Table 5 animals-12-03330-t005:** Principal component loadings and contributions to cat-initiated interaction.

Questionnaire Items	CI-PC1	CI-PC2	CI-PC3	CI-PC4
Does your cat rub its body or tail on you?	**0.77**	0.30	0.04	0.02
Does your cat purr?	**0.69**	0.40	0.11	−0.17
Does your cat call for something other than food?	**0.66**	**−0.60**	−0.14	0.16
Does your cat call to get attention from family members?	**0.64**	−0.22	−0.39	−0.09
Does your cat make eye contact with you?	**0.62**	0.04	0.25	0.08
Does your cat call to demand food?	0.29	**−0.67**	0.13	0.06
Does your cat spend time near people throughout the day?	0.20	**0.52**	**−0.58**	0.29
Does your cat knead?	0.35	−0.06	**0.54**	0.10
Does your cat respond when you call its name?	0.30	−0.12	**−0.51**	−0.48
Does your cat sleep with someone in the family?	0.19	0.10	−0.06	**0.83**
Does your cat sit on the lap of a family member?	0.39	0.37	0.35	−0.31
Eigenvalue	2.83	1.54	1.28	1.19
Contribution rate (%)	25.76	14.00	11.67	10.79
Cumulative contribution rate (%)	62.23			

Loads above |0.50| are shown in bold.

**Table 6 animals-12-03330-t006:** Correlations between hormone levels and principal component scores.

Principal Score	Cortisol (ng/mg Creatinine)	Oxytocin (pg/mg Creatinine)
	*n* = 36	*n* = 31
	r_s_	r_s_
OI-PC1 tactile and auditory interaction	0.10	0.48 **
OI-PC2 playing and brushing interaction	0.04	0.00
OI-PC3 food interaction	0.01	−0.11
CI-PC1 active interaction,	−0.20	−0.06
CI-PC2 passive interaction	−0.24	−0.08
CI-PC3 ambivalent interaction	−0.04	0.29
CI-PC4 sleep interaction	−0.24	0.17
LAPS	0.30	−0.14

** *p* < 0.01.

**Table 7 animals-12-03330-t007:** Correlations between principal component scores (OI-PC1) and other variables (individual variables, housing environment, cat personality, and owner attachment).

Variables	OI-PC1 Tactile and Auditory Interaction
	*n* = 31
	r_s_
Age (years)	−0.41 *
Body weight (kg)	−0.18
Physical activity ^1^	0.34
Number of cats	0.33
Number of family members	−0.07
Length of cohabitation (years)	−0.49 **
Neuroticism	−0.21
Extraversion	0.13
Dominance	−0.20
Impulsiveness	0.43 *
Agreeableness	0.14
LAPS	−0.30

The number of samples (*n*) for ^1^ Physical activity is 24. * *p* < 0.05, ** *p* < 0.01.

## Data Availability

The data presented in this study are available in Appendix A.

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
