# Peer review of "Physiological Assessment of the Health and Welfare of Domestic Cats—An Exploration of Factors Affecting Urinary Cortisol and Oxytocin"

_animals, 2022, doi:10.3390/ani12233330_

Round 1

Reviewer 1 Report

General comments

This study focuses on an important aspect of the relationship between domestic cats and humans, namely the effects of the inter-individual relationship on certain physiological indicators. The results of the study are important firstly because little research is carried out in the home environment, during a cat's 'normal' daily routine, and secondly because it concerns not only the effects of a relationship on an indicator of stress but also on an indicator of well-being.

The paper is well written. I only have some concerns that are the followings:

Page 8, Table 4. Why ‘How often does your cat play with you?’ is considered an owner-initiated interaction? Is the question badly formulated? Probably it should be 'How often does your cat invites you to play?' and not 'How often does your cat play with you?'

Page 8, Table 5. Concerning the question ‘Does your cat stare into your eyes?’ I have a comment: in many mammals, 'staring' into the eyes is a threatening behaviour, as much as 'looking away' is a submission/avoidance behaviour. In the domestic cat it is not quite so in the sense that staring into the eyes does not have the same meaning as for the domestic dog and for many non-human primates, but it is certainly not an affiliative behaviour. So, it is a bit strange to use it as a variable along with all the others, that are expressions of affiliative behaviour. I do not understand if the authors give another meaning to this behaviour. On the other hand, there is no definition and description of the behavioural patterns utilised in the paper (an ethogram is lacking).

Discussion

I think that sometimes, the development of the discussion of the results is too speculative. For example, there is a complex discussion on the association between the cortisol concentrations and personality traits in cats (mainly focused on highly impulsive cats), whereas in this study it was found no association between them which, as the authors themselves say, is consistent with the results of other studies. The results of this study are important because, as already mentioned, cats are rarely studied in the home environment without being subjected to invasive manipulation (in fact, urinary cortisol has not proved to be a particularly effective physiological indicator), and indicators giving a measure of cat well-being are rarely utilised, but it should not be forgotten that the only significant result of this study is a significant correlation, yes, but whose rs is 0.48, so not particularly high. Thus, excessive speculation should be avoided.

Minor points

Materials and Methods

2.4.4. Creatinine. This is the first time that you mention the creatinine utilised in order to correct variations in hormone concentrations due to urine volume. You should mention it before, both in abstract and in the introduction

Table 1. In the pdf that I have there is something funny in the ‘Response alternative’ All of them seem mixed up.

 2.7. Statistical analyses. I think that you should give the level of significance as p = … and not as p < 0.05.

Author Response

Dear Reviewer1:

We thank you and the reviewers for your thoughtful suggestions and insights. The manuscript has benefited from these insightful suggestions. I look forward to working with you and the reviewers to move this manuscript closer to publication in the Animals.

The manuscript has been rechecked and the necessary changes have been made in accordance with the reviewers’ suggestions. The responses to all comments have been prepared and attached herewith. Please see the attachment.

Thank you for your consideration. I look forward to hearing from you.

Sincerely,

Reviewer 2 Report

General comments:

This study focuses on an interesting and understudied area concerning a cat’s social environment within the domestic home, namely their social interactions with humans, and two non-invasively collected physiological measures. However, in various places, the reporting of this study, its use of literature and interpretation of results suggest a lack of critical understanding of the ethology of the domestic cat and the associated literature, in addition to a broader understanding of the limitations of psychological measures as a means to assess animal welfare. Specific comments to these effects are provided.  

Abstract:

Ln 30: more frequent in dyads of cats and owners with shorter breeding periods – I don’t understand what ‘short breeding periods’ means in this context

Introduction:

ln 42: and the state of attention and interest in themselves from a persons gaze – the sentiment in this sentence could benefit from some rewording for clarity

ln 43: Cats can identify their owners' voices [8]. – would be beneficial for the reader to add a little context/explanation here

ln44: This suggests that cats recognize their owners as special – the word special seems a little subjective and anthropomorphic – suggest using different terms here

ln45: Social interaction with owners is essential for improving cat welfare [9] – at a species level this is absolutely not true - for unsocialised or unsociable cats the opposite argument is much more likely to be the case. Additionally, even amongst socialised pet cats, there is evidence to suggest that petting and inappropriate social interactions with humans can induce negative affective states/aggression

For example, 45 contact with people reduces the risk of respiratory diseases in cats living in shelters [10– 46 12]. – the associations are more nuanced than this – in these studies, these health benefits were only identified in cats that were considered to respond positively to human interactions. Please avoid making sweeping statements about the welfare benefits to cats from humans interactions with them, because this is anthropocentric and does not translate to good welfare messages for cats at a general population level.   

Ln 47: Human interactions alter the physiological and psychological state of cats and de- crease cortisol levels, which reflect their stress state and related behaviors [13] – I find this statement a little problematic for the reasons mentioned above. Also, there is a wealth of multi-species literature including that focused on the domestic cat which highlights the various limitations of using physiological measures such as cortisol to make inferences about the welfare and particularly emotional valence of an individual - it’s important that this is properly acknowledged and with appropriate caveats included.  

Ln49: and is essential for promoting feline health – this sentence seems a little vague and I’m not sure logically/meaningfully fits here

Ln 50: However, methods to quantify the physiological 50 health and welfare status of cats, especially those living in general households, are limited, 51 and few studies have been conducted to analyze this [15] because cats are solitary hunters 52 [16] and do not undergo a transparent domestication process like dogs [17], resulting in 53 highly independent behavioral characteristics. This indicates that sufficient training is re- 54 quired for collecting blood and saliva samples from domestic cats invasively. Further- 55 more, cats are territorial animals [16] and, the visit of an researcher induces a strong 56 stress response in them– suggest rewording of this section to improve logic and clarity of arguments

Ln 57: collection of hair and especially nail samples are not inherently non-invasive if they involve restraint and induce stress

Ln 58: . In the 58 aforementioned studies, cortisol, a stress hormone, was measured to assess the physiolog- 59 ical stress states in cats. However, to improve animal health and welfare, it is essential to 60 not only decrease the negative factors but also to increase positive factors. – I find this sentence problematic as it again seems to be based on the premise that physiological arousal is directly and unidirectionally correlated with negative emotional arousal, and this is not the case.

Ln68: this would seem an appropriate place to mention a previous paper published by the authors (Nagasawa et el 2021) which appears to present results contradictory to those focused on the impacts of HAI in dogs which are being mentioned here  

Ln 77: Therefore, there is an urgent need of research focusing on  the oxytocin levels in the owner-cat relationship [39] – I would suggest phrasing this a little more objectively and conservatively – no ‘urgent need’ has been demonstrated within the arguments presented

Ln 79: In this study, we aim to assess the health and welfare status of cats living in general 79 households.- this is not technically accurate – the study investigates two physiological measures that are associated with aspects of health and welfare, the study is not measuring the ‘health and welfare status’ of cats in any global or holistic way as this sentence suggests

Methods:

Ln91: (n = 49; male = 24 and female = 25; mean age = 5.30 ± 4.06) – it is unclear whether these values refer to people or cats!

Ln92: suggest some rewording for clarity, also is there any evidence that this is an appropriate time frame?

Ln103: what substrates and type of litter boxes were used to collect the urine in? if these were different to those owners usually provided to cats, were cats given any time to habituate to these new tray provisions? what were the time frames from collection to freezing to being stored at the lab? Was urine stored in lab freezers for any length of time and if so, at what temperatures? Were owners instructed to collect samples at different times each day or the same times?

Ln 134: what does ‘breeding period’ mean?

2.5: it’s great so see owner-initiated and cat-initiated interactions distinctly separated here as these likely present important distinctions in the nature and quality of human-cat interactions

Table 1: some items seems to be repeated several times at different points in this table – are these typos?

Ln 140: was the feline five competed by owners? From the current wording this is a little unclear

Ln 148: please indicate what constituted a ‘considerable stress response’

Ln 153: The mean concentrations of cortisol and oxytocin per individual were calculated from total urine samples per individual – what does ‘total urine samples’ mean?

Ln157: please provide sufficient information regarding your PCA – what tests were performed to determine if your data were suitable to undergo a PCA? What type of PCA rotation was applied? How was the PCA and it’s components used to create human-cat interaction scores?!

If multiple cats were sampled from the same home in some cases, this should be factored into the nature of your statistical analysis

Results:

The mean urinary cortisol concentration of 36 cats was 3.83 ± 1.63 ng/mg creatinine. The mean urinary oxytocin concentration in 31 cats  - please clarify why only 36 cats were sampled for cortisol and why 31 for oxytocin.

Discussion:

Ln 209-222 – this seems to be largely a repetition of information provided in the introduction and I would suggest doesn’t add value to the discussion as it is currently presented

Ln 213: and is an essential indicator for assessing the health and welfare of animal species – I disagree that there is sufficient/compelling evidence to suggest that it is an ‘essential’ indictor and would suggest a more conservative term is used here such as valuable or important

214 However, no study had investigated  oxytocin in cats – given the authors nice previous work on oxytocin in cats in find this sentence a bit confusing!

Ln 227: Our results suggest that many owners influence the welfare status of their cats through routine communication -  oxytocin is not synonymous with a cat’s ‘welfare status’ – please use more conservative wording when referring to the relevance of urine oxytocin concentrations to a cat’s overall wellbeing

Ln 230: Cats perceive human petting and hugging as allo-grooming and allorubbing by their owners – this is an anthropocentric statement and is not supported by any empirical evidence that I’m aware of, please remove or rephrase to something more biologically and scientifically appropriate

Ln 232: suggesting that they are extemely interested in vocal stimuli. – check spelling. I don’t think ‘extremely interested’ is the right term to use here as it is implying motivations and perspectives in cats than have not been explicitly tested in the research being referred to. ‘sensitive to’ would be a more appropriate term

Ln 232: Thus, tactile and au-ditory communication from the owner is an important signal and factor associated with the physiological state of the cat. – please see previous comments about generalisation and the technical/biological accuracy of arguments and reword this statement accordingly

Ln 239-246: Previous studies have found that tactile and auditory stimulation from 239 mothers profoundly affects the health status of infants….. – in what species?

L246: Cats develop an attach-ment to their owners [51,52] and form social relationships similar to those between parents  and children. – this is debatable given that the most methodologically sound paper that tested this hypothesis (Potter and mills 2015) provides contradictory evidence to this

Ln 248: Human voice and contact stimulation have been found to decrease stress 248 and improve immune function in cats, thereby reducing the risk of upper respiratory dis- 249 eases [10–13] – please see my previous comments regarding how these papers are being referenced and the potentially poor welfare messages if this is not discussed in a more nuanced and accurate way

Ln255: greater 256 frequency of contact and vocal interactions (Table 7). – please change to ‘of owner initiated’ contact…

Ln 257: In other words, we hypothesized 257 that cats actively communicated with their owners when the social relationship between 258 them was still developing – this logically doesn’t seem to make sense given that the results being referred to relate to OI-PC1 which is owner initiated rather than cat initiated social interactions  

section commencing ln 261:

Several key studies are missing from this discussion including one previously published by the authors. Perhaps they do not support the narrative within the paper, but they are important to discuss for a more objective perspective of oxytocin in cats and its associations with human-cat interactions and

Arahori, M.; Chijiiwa, H.; Takagi, S.; Bucher, B.; Abe, H.; Inoue-Murayama, M.; Fujita, K. Microsatellite Polymorphisms Adjacent to the Oxytocin Receptor Gene in Domestic Cats: Association with Personality? Front. Psychol. 2017, 8, 2165.

Nagasawa, T.; Ohta, M.; Uchiyama, H. The Urinary Hormonal State of Cats Associated with Social Interaction with Humans. Front. Vet. Sci. 2021, 8, 680843. [CrossRef]

Ln 265: The genes 265 of the oxytocin and arginine vasopressin systems in cats, which have functions in social 266 behavior, aggression control, and stress processing, have changed compared to their wild 267 ancestor [58], and thus cats form attachment relationships with their owners [51,52] – unsure of the logic of this sentence. Please also see previous comments regarding attachments in cats

Ln 270 Our results are the first to suggest that the oxytocin system is involved in forming cat-owner relationships. – I’m not sure that this is true, see Arahori et al 2017

Ln280 Cat 280 personality is an essential factor for assessing the welfare status of cats [59,60] – this doesn’t make sense

Ln290-300 Please see previous comments regarding the limitations of cortisol as a measure of stress/negative welfare states and associated with behavioural indicators

Ln307: any evidence for the lag between stress exposure and secretion of stress hormones in urine?

Ln 320: . Oxytocin is thought to be involved in the attachment be- 320 tween owners and cats -  is this speculative or evidence supported?

Ln 333: in conclusion, urinary oxytocin levels in cats living in general households are associ- 333 ated with the frequency of contact and vocal communication with their owners. – please clarify that this is owner directed to the cat not the other way round

Ln 334 In other words, our results suggest that positive interactions with owners are essential for improving the health and welfare of cats. – this conclusion is not supported by the current study – please remove

Ln336: Owner-cat dyads with shorter cohabitation periods had 336 more frequent contact and vocal communications. – see previous comment re direction of social behaviour

Author Response

Dear Reviewer2:

We thank you and the reviewers for your thoughtful suggestions and insights. The manuscript has benefited from these insightful suggestions. I look forward to working with you and the reviewers to move this manuscript closer to publication in the Animals.

The manuscript has been rechecked and the necessary changes have been made in accordance with the reviewers’ suggestions. The responses to all comments have been prepared and attached herewith. Please see the attachment.

Thank you for your consideration. I look forward to hearing from you.

Sincerely,
